# Some aspects of the deep abyssal overflow between the middle and southern basins of the Caspian Sea

# 5 Javad Babagoli Matikolaei<sup>1</sup>, Abbas Ali AliAkbbari Bidokhti<sup>2</sup>, and Maryam Shiea<sup>3</sup>

<sup>1</sup>Graduate in Physical Oceanography, Institute of Geophysics, University of Tehran, Tehran, Iran.
 <sup>2</sup> Institute of Geophysics, University of Tehran, Tehran, Iran.
 <sup>3</sup> Faculty of Marine Science and Technology, Science and Research Branch, Islamic Azad University, Tehran, Iran.

Correspondence to: J. Babagoli Matikolaei (javadbabagoli@ut.ac.ir)

Abstract. This study investigates the deep gravity current between the middle and southern Caspian Sea basins, caused by density difference of deep waters. Oceanographic data, numerical model and dynamic models are used to consider the structure of this Caspian Sea abyssal overflow. The CTD data are obtained from UNESCO, and the three-dimensional ocean

- model COHERENS results are used to study the abyssal currents in the southern basin of the Caspian Sea. The deep overflow is driven by the density difference mainly due to the temperature difference between the middle and southern basins especially in winter. For this reason, water sinks in high latitudes and after filling the middle basin it overflows into the southern basin. As the current passes through the Absheron Strait (or sill), we use an analytic model for the overflow gravity current with inertial and frictional effects to consider its structure. The dynamical characteristics of this
- deep baroclinic flow are investigated with different initial and boundary conditions. The results show that after time passes, the flow adjusts itself, moving as a deepening gravity driven topographically trapped current. This flow is considered for different seasons and its velocity and width are obtained. Because of the topography of the Southern Caspian basin, the flow is trapped after the sill; thus, another simple dynamical model of the overflow, based on potential vorticity conservation similar to that of Bidokhti and Ezam (2009) but with the bottom friction included, is used to find the horizontal extent of the
- outflow from the western coast. The result of this model shows that the Rossby length (deformation radius) of the flow decreases when drag coefficient increases. Because of the importance of the overflow in deep water ventilation, a simple dynamical model of the boundary currents based on the shape of strait is used to estimate typical mass transport and flushing time which is found to be about 15 to 20 years for the southern basin of the Caspian Sea. This time scale is important for the possible effects of pollutions due to oil exploration on the ecosystem of this water body.
- Keywords: Overflow, dynamical model, trapped baroclinic bottom current, Caspian Sea abyssal flow.

# 1. Introduction

Baroclinic flows play important roles in ocean and sea circulations, especially in deep waters of the ocean. Because these currents are important in deep water ventilations of the oceans, they have an integral role in thermocline circulation. Vertical ocean circulation that is created by density differences and controlled by changes in surface temperature and salinity is known as thermohaline circulation. A driving mechanism for the circulation is the cooling of surface waters at high latitudes

- and consequent formations of deep waters by sinking the cooled salty water masses (Fogelqvist et al., 2003). Cooling in polar seas (Dickson et al., 1990) and evaporation in marginal seas (Baringer and Price, 1997) cause dense waters that sink to form deep water masses. For example, dense water from the deep convective regions of the North Atlantic produces a signature of the thermohaline overturning circulation that can be seen as far away as the Pacific and Indian
- oceans (Girton et al., 2003). In the global sense, bottom-trapped currents play an integral role in thermohaline circulation and are a vehicle for the transport of heat, salt, oxygen and nutrients over great distances and depths. The ability of abyssal flows to transport and deposit sediments is also of geological interest (Smith, 1975). As thermohaline circulation causes ventilation of deep ocean water, it is important not only in open seas and ocean but also in semi-closed and closed basins ventilations. Study of thermohaline dynamics and circulation has been of interest to different scientists as climate researchers. The
- dynamics of such dense currents on slopes have been modelled in the past both theoretically and experimentally starting with Ellison and Turner (1959) and Britter and Linden (1980), and a review on gravity currents can be found in Griffiths (1986). Bidokhti and Ezam (2009) presented a simple dynamical model of the outflow from the Persian Gulf based on potential vorticity conservation to find the horizontal extent of the outflow from the coast.
- The Caspian Sea, the world's largest inland enclosed water body, consists of three basins namely northern (shallow, mean depth of about 10 m and covering 80000 km<sup>2</sup>), the middle (rather deep, with mean depth of about 200 m, maximum depth 788 m and covering 138000 km<sup>2</sup>) and the southern (deep, with a mean depth of 350 m, maximum depth 1025 m and covering 164840 km<sup>2</sup>) and is located between 36.5 N and 47.2N, and 46.5E and 54.1E (Aubrey et al., 1994; Aubrey, 1994). The bathymetric levels vary largely over this sea (Ismailova, 2004, Figure 1). The northern basin, after a sudden depth transition at the shelf edge, reaches the middle one. The middle and southern basins are divided by the Absheron sill or strait
- (with maximum depth of 180 m). The western slopes of the two deeper basins are fairly steep compared to the eastern slope (Gunduz and Özsoy, 2014).

The main aims of our work are interpretations and discussions of the abyssal flows in the southern Caspian Sea basin. For this reason, we used CTD data and simulations of a numerical model. Firstly, we explain why abyssal circulation occurs in the Caspian Sea, and then discuss the structure of the outflow (overflow) from the middle to the southern Caspian Sea over

30 the Absheron sill. In this study an analytical model for the overflow gravity current over the Absheron sill, with inertial and frictional effects, is then presented. After the flow passes over the Absheron sill, we present a new model for the structure of this flow using another analytical model based on potential vorticity conservation, similar to that of Bidokhti and Ezam

(2009) but the bottom friction is included. We then calculate values of mass transport, flushing times, and discuss the pathway of gravity current in southern Caspian Sea basin.

# 2 Data used and results

# 5 2.1 Observational data

The CTD data are obtained from UNESCO Atomic Energy International Agency, for summer 1996 (Figure 1). Vertical profiles of temperature, T, salinity, S and sigma-T, for the northern, middle, and southern Caspian Sea basins, show differences in temperature, salinity and sigma-T between these basins. Figure 2 shows typical vertical profiles of T, S and sigma-T in these basins (points a and b in Figure 1 right). Due to the density difference between the middle and southern basins, dense current crosses the Absheron sill. The cross-section of sigma-T shows that the abyssal gravity current moves

- 10 basins, dense current crosses the Absheron sill. The cross-section of sigma-*T* shows that the abyssal gravity current moves from middle to southern basin (Figure 3). Not only in the Strait of Absheron are sigma-*T* contour gradients obvious but also there are contour gradients in southern basin, indicative of the northern overflow. Due to the limited observational data, more extensive measurements and high resolution data are required to find the outflow structure and determine the analytical model parameters. For this reason, numerical model results have been used to do further analyses and also acquire the
- 15 parameters of the overflow for the analytical model.

Figure 1: Scheme of the Caspian Sea and locations of CTD measurements, the geographic position of transects (right), and topography of the basins, showing the Absheron sill (left).