# Peer review of "Some aspects of the deep abyssal overflow between the middle and southern basins of the Caspian Sea"

_Ocean Science, 2016_

## Referee Comment (RC1) · Anonymous Referee #1 · 23 Feb 2017

Although perhaps not pure "Ocean" science, the subject of deep overflow between two basins of the Caspian Sea ought to be of sufficient interest to the readers of Ocean Science to merit publication. Unfortunately, I do not find that this manuscript presents significant new information on that subject and it seems to contain highly questionable results, as will be elaborated below. I therefore cannot recommend publication of the manuscript.

1. Not being familiar with the Caspian Sea, one of my first questions was: "What is the evidence for an overflow?" I did not find a very satisfactory answer in the manuscript. The introduction gives some general information on the Caspian Sea, but I missed information on what is known about an overflow. Has it been discussed in previous

scientific publications and what is the evidence? I only found a reference to Gunduz and Özsoy (2014), in which an overflow is mentioned, but there, they claim "limited evidence". The Introduction ought to state clearly, whether this overflow has been described in the literature and its main features. If its not been well described previously and its evidence is limited, then this should also be clearly stated in the Introduction. If not clearly identified in previous studies, observational evidence for an overflow ought to be presented, but the only observational evidence for an overflow that I found in the manuscript was in Sect. 2.1, where CTD profiles from two stations were compared and it is stated that "Due to the density difference between the middle and southern basins, dense current crosses the Absheron sill". This is indeed a possibility, but certainly not the only one. And, even if there is an overflow, how deep does it go? Is it persistent or intermittent? etc.

2. Thus, the only evidence for an overflow, that I found in the manuscript, is from modeling and the manuscript presents results from three separate models, one numerical and two analytical models. I have no experience with the COHERENCE numerical model, but offhand the results presented seem interesting. I miss, however, information on some key questions: How well does the model represent the (apparently scarce) observational evidence, such as the CTD profiles ? How sensitive is it to mixing parameters ? Does it show overflow ? Especially the last question would be very relevant. From Figure 4, the temperature close to the bottom of the southern basin is not equal to that at the sill. Does this mean that the overflow is modified by entrainment on route ? Or that it does not go all the way to the bottom ? Or is there a well-defined overflow in the model and what are its properties ? The manuscript refers to Figure 4 and states that: "...the overflow over the Absheron sill and in the north western boundary of the southern basin are clearly observed", but I do not see this clearly from the figure. Yes, there is strong flow, but is it descending and is the descent density-driven ? I miss this information in the text. It appears that the numerical model is mainly used to provide parameters for the analytical models.

3. The first analytical model (Sect. 3.1) looks to be a very simple model. I have not checked the equations and will assume that they are valid, but I find the definitions of the model badly presented. The text says that: "local coordinates xÊź is along the flow", but if that were the case, you would expect the y-velocity v' to be identically zero. From the equations, I assume that the bottom is a flat plane tilting an angle $\theta$ from the horizontal and that the x' direction is downslope. But, how well can that represent the conditions at the Strait of Absheron. In Figure 9, the model tracks a water particle moving 30 km downslope, which must imply a deepening of 30000x0.02 = 600 m. On its way, it has to pass transect C, where the dense water according to Figure 6 is not on a plane, but confined within a fairly narrow channel. Also, the model seems internally inconsistent. In Figure 1, I have plotted the tracks of two water particles starting downslope at the same time, but with one starting 5 km displaced relative to the other (in y-direction). The two particle tracks cross. Apparently, this model describes an individual water particle, not affected by the motion of the neighbouring water, and I do not see any reason that it should reflect reality.

4. The second analytical model also seems very questionable. It assumes conservation of potential vorticity, but it includes friction and PV-conservation usually assumes inviscid flow. The text says: "... the potential vorticity is conserved, because of topography in the South Caspian", which is not clear to me. But, there are other problems with the model. From the definition of h, Eq. (6) is for the upper layer, but Eqs. (7) and (8) must be for the lower layer. How can they be combined ? Like in the previous model, friction is assumed to depend linearly on velocity, which is not very realistic and there is no justification for the chosen values for r.

5. The results section ends with a calculation of flushing time, but again I have difficulties with understanding it. Where does the expression for h in Eq. (13) come from ? It is not consistent with Eq. (10). When it is stated that L1 = L2 = L, I assume that there is a minus sign missing, but even so, these points should be where the interface hits bottom on both sides of the channel (Figure 12). How can you assume that they

are symmetric around the center of the channel ? There is a rich literature on analytical overflow models, even with parabolic bottom shape (e.g., Borenäs and Lundberg, 1988). I would suggest using one of those for this calculation.

6. There are a number of details that should be corrected: small figures, small figure legends, using rho instead of sigma in some figures (is it corrected for pressure ?), using the term "Rossby length" to represent the actual width of the flow, the reference to Gunduz and Özsoy (2014) in the reference list is wrong.

[Figure]

[Figure]

**Fig. 1.** Figure 1. Tracks of two water particles (y' versus x'), starting at time t=0 with the red track displaced 5 km from the blue using Eq. (3) for NOV in Table 1 and r=0.00001 s-1.

---

## Short Comment (SC1) · 14 Mar 2017

Our answer and further clarifications are as follows:

As mentioned in the paper, not much work on the deep flows and waters of the Caspian Sea basins, the largest water body in the world have been done, hence, the present work may be the first attempt to concentrate on such issues which are crucial for the future fate of this sea, as have been addressed in the papers conclusion section. Although there are shortcomings in the paper, particularly direct observations of deep currents in the Caspian Sea, however, I am surprised to hear that there is not much NEW in the paper by the referee #1!

There are two important points concerning the importance of this research. The first point is that there are a large number of evidences which show signs of life in the depths of the Caspian Sea, especially in the Southern basin of Caspian Sea. For example, you can observe this signs in below Scheme (**Biological Features and Resources Ocaspian Sea**, M.G. Karpinsky  $\cdot$  D.N. Katunin  $\cdot$  V. B. Goryunova  $\cdot$  T.A. Shiganova, In Caspian Sea Environment, Spinger, Part P (2005): 191–210).

| Depth | Oxygen | Phyto-
plankton | Zoo-
plankton | Benthos | Fishes |
|-------|--------|--------------------|------------------|---------|--------|
| 200   |        | 7                  | T                |         | T      |
| 400   |        |                    |                  |         |        |
| 600   |        |                    |                  |         |        |
| 800   | 41111  |                    |                  |         |        |
| 1000  |        |                    |                  |         |        |

Fig1. Scheme of vertical distribution of oxygen phytoplankton, zooplankton benthos, and fishes.

Which means that the Caspian Sea is not as Black sea which is due to lack of such ventilation is nonproductive in deep parts. But you cannot also find any research to answer this question as why the deeper part of the Caspian Sea is rather alive. The current study shows that such abyssal flow can ventilate the deeper parts of the Caspian Sea.

Another point is oil pollution due to oil exploration, especially in the Strait of Apsheron, which is under way that can affect the deep parts of the southern basin of this Sea (see Fig 2).

**Fig2.**

1. The density or better sigma (will be corrected in the paper) fields of the two middle and southern basins inferred by the direct observations indicate that the middle basin, especially in the deeper parts is denser than the southern one (as an example of density profiles, corrected for pressure effect, is shown in the paper). This is also expected as the higher latitude of the northern and middle basins are more prone to be exposed colder and stronger westerly winds and mid-latitude cyclones and cold fronts than the southern basins, the northern basin also freezes in the cold season (a figure of measured monthly mean air temperature over the basins will be added to the paper, fig. 4). For this purpose we used meteorological data and MODIS data in Caspian Sea for three positions of this points in Fig3.

---

## Referee Comment (RC2) · J. M. Huthnance (Referee) · 18 Apr 2017

This manuscript has already received one review with severe criticisms. Two more reviewers were nominated but the reviews have not yet been received. I think the comments of the first referee should be addressed. In my capacity as editor I expect to refer any revised version to at least two of the referees for re-consideration. This comment is to expedite the review process.

---

## Editor Comment (EC1) · J. M. Huthnance (Editor) · 19 Apr 2017

Dear Authors If you are confident (as I think) that you are answering the comments of Referee 1 (you may agree or disagree with Referee 1) then please do revise your manuscript. Your answers should be merged into the revised manuscript so that the text reads well and addresses the referee 1 comments. I still expect to send it back to two referees. Yours sincerely John Huthnance

---

## Author Comment (AC1) · 19 Apr 2017

Our answer and further clarifications are as follows:

As mentioned in the paper, not much work on the deep flows and waters of the Caspian Sea basins, the largest water body in the world have been done, hence, the present work may be the first attempt to concentrate on such issues which are crucial for the future fate of this sea, as have been addressed in the papers conclusion section. Although there are shortcomings in the paper, particularly direct observations of deep currents in the Caspian Sea, however, I am surprised to hear that there is not much NEW in the paper by the referee #1!

There are two important points concerning the importance of this research. The first point is that there are a large number of evidences which show signs of life in the depths of the Caspian Sea, especially in the Southern basin of Caspian Sea. For example, you can observe this signs in below Scheme (**Biological Features and Resources Ocaspian Sea,** M.G. Karpinsky · D.N. Katunin · V. B. Goryunova · T.A. Shiganova, In Caspian Sea Environment, Spinger, Part P (2005): 191–210) .

[Figure]

**Fig1**. Scheme of vertical distribution of oxygen phytoplankton, zooplankton benthos, and fishes.

Which means that the Caspian Sea is not as Black sea which is due to lack of such ventilation is nonproductive in deep parts. But you cannot also find any research to answer this question as why the deeper part of the Caspian Sea is rather alive. The current study shows that such abyssal flow can ventilate the deeper parts of the Caspian Sea.

Another point is oil pollution due to oil exploration, especially in the Strait of Apsheron, which is under way that can affect the deep parts of the southern basin of this Sea (see Fig 2).

[Figure]

**Fig2.**

1. The density or better sigma (will be corrected in the paper) fields of the two middle and southern basins inferred by the direct observations indicate that the middle basin, especially in the deeper parts is denser than the southern one (as an example of density profiles, corrected for pressure effect, is shown in the paper). This is also expected as the higher latitude of the northern and middle basins are more prone to be exposed colder and stronger westerly winds and mid-latitude cyclones and cold fronts than the southern basins, the northern basin also freezes in the cold season (a figure of measured monthly mean air temperature over the basins will be added to the paper, fig. 4). For this purpose we used meteorological data and MODIS data in Caspian Sea for three positions of this points in Fig3.

[Figure]

**Fig3.**

[Figure]

**Fig 4.** Comparison the air temperature in North (blue), Middle (green) and South (red) Caspian Sea.

As shown in the northern basin, the temperature can reach -8°c in winter. In the next step, we plot time series of surface temperature and density changes for points shown in Fig 3 using MODIS data.

[Figure]

(a)

(b)

**Fig 5.** Comparison the surface temperature and sigma-T in North (blue), Middle (green) and South (red) the Caspian Sea.

Another important evidence about the overflow is the numerical model output. For example, you can see the overflow in the cross-section along the Latitude in fig. 6.

[Figure]

**Fig 6**. Cross-Section of the mean temperature obtained from model simulation during Febuary and July.

Hence, overflow from the middle to southern basin is expected, although direct measurement of this flow has not been carried out. This is why we used numerical simulations which have been verified against some existed measurements only in the southern basin that will be added to the paper (fig. 7). The overflow certainly has seasonal changes which we have also indicated and have been taken into account in the paper, but its detail possible intermittent behavior should be the subject of another work. The introduction will also be amended accordingly, as pointed out be the referee #1, although as was mentioned before not much work have been done on this overflow.

2.  About validation of numerical model, yes we validate the results of numerical model by CTD and ADCP data. For this reason, we run this model in certain years to compare it results with observation data. For example, you can see ccomparison between the numerical model results and observation in Fig 7, near Bandar-e Anzali (long.: 37.5°, lat.: 50.5°), indicating rather close agreements.

[Figure]

**Fig7**. Comparison between numerical model results of current components and observation.

About temperature differences in the Strait and the South Caspian Sea, it should be noted that model results for Sigma coordinate have been presented so, depth is not equal in each k along the flow. The actual depths of some points along the flow in the Caspian Sea are shown in Fig 8. For this reason, temperature is not the same in the Strait and parts of South Caspian, see also Fig 6.

[Figure]

**Fig 8.**

Yes, as measurements are scarce (mentioned in the paper) we used some of the numerical results to provide parameters for the analytical mode. Yes mixing parameterizations, within the model limitations, are included with proper turbulence scheme as Miller-Yamada in numerical simulations. The flow descends as a density driven (as g' the reduced gravity of the over flow is negative, hence down ward flow) current then is trapped by the side boundary of the basin, due to rotational effect (Fig 6). Due to bottom friction it should spiral down the basin as indicated to some extent by the numerical results.

3.  The choice of the coordinates for the first analytical model may not be as popular, but the model is an initial condition Lagrangian one without entrainment (or interaction of fluid particle with the surrounding ones, a shortcoming for this stage), hence yes, it only shows the path of the fluid particles. The bottom slope of model was also estimated from the topography of Absheron sill. About the path of two particles, a particle cannot have an effect on another particle because we ignore the effects of stirring diffusion. If we consider this effect, we cannot present such an analytical model for this flow which is time non-steady and three dimensional, but there is some paper deal with Stirring diffusion only for a steady case (see Cenedese, C. J. and Whitehead, A., 2002. "A dense current flowing down a sloping bottom in a rotating fluid", J. Phys. Oceanogr, Vol. 34, 188-203). We had to choose either steady with stirring diffusion or non-steady without diffusion to solve momentum equations, as the path of the flow was important.
Concerning particle shift as much as 30 km, we should note that a particle cannot move about 30 km in South Caspian due to topography, while sinking down. In Fig 9 (paper) we just show that the path of flow in the absence of topography. As θ is constant, in reality the flow move about 10 km (maximum) as the strait is short, then the flow is trapped by topography and does not find the opportunity to do oscillating motion (Fig 7 in paper, or the below figure).

[Figure]

**Fig 9.**

As seen in Fig 10, flow moves about 8.5 km, then with your estimation the flow descent only about 170 meters (8500×0.02=170 m).If you note the numerical model results (Fig 10), flow descent about 140 or 150 meters. It is logical because of our assumption (with no diffusion).

[Figure]

**Fig 10.**

Concerning accuracy of this model, you can see Appendix A.

4. In this analytical model, we use conservation of potential vorticity (PV) and assume that vorticity of flow is $\xi_1$ in the strait of Apsheron (Fig 11) and $\xi_2$ ($=\xi_1+c$) when the flow is trapped. We consider c=0, so potential vorticity is conserved. We have to consider c=0 because it is impossible to estimate c. In last article (Bidokhti and Ezam) $\xi_1$ is divided into two vorticities, relative and planet vorticity, because they consider geostrophic flow. In this paper, we write momentum equation based on Quasi-geostrophic flow. In this situation, the angle between the pressure gradient and Coriolis is not 90 degree, due to bottom friction which was considered in our model (please see Fig 12).

[Figure]

**Fig 11.**

[Figure]

**Fig 12** A model of steady trapped flow affected by bottom topography (Girton, J. B., Sanford, T. B., 2001. "Descent and Modification of the Overflow Plume in the Denmark Strait", Geophys. Res. Lett., 28, 1619–1622).

In this model, part of vorticity (relative) is converted due to friction. For this reason when we increase r in the model Rossby deformation radius decrease because part of relative vorticity is more converted or the angle between the pressure gradient and the Coriolis is further reduced. It is not against the Conservation of potential vorticity because $\xi_1 \approx \xi_2$. Vorticity is distributed between three factors unlike in the case with the case with perfect PV conservation as has been used in the past.

Here we have used the model based on two layers and considered a isopynal line after the strait in the South Caspian then, we write momentum equation based on this isopynal line. In this model two layers move together and we can define boundary conditions (BC) better when consider the surface BC. This method is common in other paper, e.g. please Bidokhti, A. A. and Ezam, M., 2009. The structure of the Persian Gulf outflow subjected to density variations, Ocean Sci., 5, 1–12, doi: 10.5194/os-5-1-2009).

In this paper, coefficient of friction is a linear function of velocity. Some paper assume linear for example see this equation:

$$-r\underline{u} = \frac{rg'}{f} \underline{k} \times \nabla(H - h) - \frac{r^2}{f}\left(\underline{k} \times \underline{u}\right).$$

In Mansley, H.C., and Marshall, D., 2004. "Dense Water Overflows and Cascades", University of Reading, pp.1-65.

5- In this article the following equation is used to calculate volume

$$Q_v = \int v dA$$

[Figure]

**Fig 13.**

To calculate velocity and area which water enter the south Caspian Sea, we use the shape of the reference isopycnal when the Strait topography crosses the isopycnal at two points. We use geostrophic flow ($v{=}g'\frac{\partial h}{\partial x}$) to calculate velocity, $v$ which means that:

$$\frac{g'}{f}\frac{H_1 - H_2}{L_2 - L_1}$$

To calculate dA, we integrate the isopynal while using the parabolic equation of the strait topograpgy, then calculated area is:

$$Q_V = \frac{g'}{f}\frac{H_1 - H_2}{L_2 - L_1}\left[\left(-\frac{A}{\alpha}(e^{-\alpha L_2} - e^{-\alpha L_1})\right) - \frac{a}{3}(L_2^3 - L_1^3) - \frac{b}{2}(L_2^2 - L_1^2)\right] \quad (1)$$

At first, we use $L_1$ and $L_2$ because the shape of topography is approximately symmetric, hence L1≈L2. Please see the following Fig

[Figure]

**Fig 14.**

To simplify, we can also assume L1=L2=L which L=(|L2|+|L1|)/2 , So we earn the following equation:

$$Q_V = \frac{g'}{f} \frac{H_1 - H_2}{L_2 - L_1} \left[ \frac{2A}{\alpha} \sinh(\alpha L) - \frac{2}{3} a L^3 \right] \qquad (2)$$

Now, we show that this assumption does not produce large differences. For example in January L2|+|L1|=32000 m we assume that under the worst conditions |L_2|=17000 m, |L_1|=15000 m and L=16000.

We calculated $Q_V$ by (1) and (2) formula above, and compare the results with each other in following table, which are not much different.

| Formula | a | b | α | A | Qv(SV) |
|---------|---|---|---|---|--------|
| 1 | 4.43×10⁻⁷ | −9.877×10⁻⁴ | 1.669×10⁻⁵ | 109.18 | 0.109 |
| 2 | 4.49×10⁻⁷ | −1.875×10⁻³ | 1.669×10⁻⁵ | 111.018 | 0.115 |

Concerning other formula to calculate $Q_V$, you suggested that we use this formula (Borenäs and Lundberg, 1988). We found the following relations.

$$Q = \frac{h^2}{2+r} \sqrt{\frac{3f^2}{2r}}$$

$$r = \alpha f^2 / g'$$

in Wilkenskjeld, S. and Quadfasel, D.,2005. "Response of the Greenland-Scotland overflow to changing deep water supply from the Arctic Mediterranean", GEOPHYSICAL RESEARCH LETTERS, VOL. 32, L21607.

The equation is presented to calculate $Q_V$ in the **Faroe Bank Channel** based on field observations in **Faroe Bank**, which seems useful to estimate $Q_V$ for that area. They used this formula to calculate the maximum volume in the current case. The values for the two method are shown in the following table, indicating that their formula gives much larger values giving the flushing time of about 10 years which will be added to the present paper.

|  | Borena's and Lundberg-$Q_V$(SV) | Current paper- $Q_V$(SV) |
|---|---|---|
| NOV | 0.028 | 0.016 |
| JAN | 0.42 | 0.115 |

In the paper table 3 L should be changed to 2L.

|  | $H_1$ (m) | $H_2$ (m) | *2L* (m) | $Q_V$ (Sv) |
|---|---|---|---|---|
| NOV | 55 | 10 | 19000 | 0.016 |
| JAN | 145 | 85 | 32000 | 0.115 |
| MAY | 145 | 55 | 34000 | 0.146 |
| SEP | 135 | 45 | 27500 | 0.116 |

Concerning accuracy of this model, you can see Appendix B.

6. The editing error in the text will be corrected, while noting the mentioned ones by the Referee #1

**Appendix A**

Derivation of the first analytical model:

$$\frac{du'}{dt} = -g'\tan\theta + fv' - ru'$$

$$\frac{dv'}{dt} = -fu' - rv'$$

$$u'(t) = c_1 e^{-rt}\sin(ft) + c_2 e^{-rt}\cos(ft) - \frac{g'r\tan\theta}{f^2 + r^2}$$

$$v'(t) = c_1 e^{-rt}\cos(ft) - c_2 e^{-rt}\sin(ft) + \frac{g'f\tan\theta}{f^2 + r^2}$$

$$x'(t) = \frac{fc_2 - rc_1}{f^2 + r^2} e^{-rt}\sin(ft) - \frac{fc_1 + rc_2}{f^2 + r^2} e^{-rt}\cos(ft) - \frac{g'r\tan\theta}{f^2 + r^2}t + \frac{fc_1 + rc_2}{f^2 + r^2}$$

$$y'(t) = \frac{fc_1 + rc_2}{f^2 + r^2} e^{-rt}\sin(ft) + \frac{fc_2 - rc_1}{f^2 + r^2} e^{-rt}\cos(ft) + \frac{g'f\tan\theta}{f^2 + r^2}t + \frac{rc_1 - fc_2}{f^2 + r^2}$$

$$\frac{dx'}{dt} = \frac{fc_2 - rc_1}{f^2 + r^2}\left[-re^{-rt}\sin(ft) + fe^{-rt}\cos(ft)\right] - \frac{fc_1 + rc_2}{f^2 + r^2}\left[-re^{-rt}\cos(ft) - fe^{-rt}\sin(ft)\right] + \frac{-g'r\tan\theta}{f^2 + r^2}$$

$$u' = \frac{fc_2 - rc_1}{f^2 + r^2}e^{-rt}(f\cos(ft) - r\sin(ft)) + \frac{fc_1 + rc_2}{f^2 + r^2}e^{-rt}(r\cos(ft) + f\sin(ft)) + \frac{-g'r\tan\theta}{f^2 + r^2}$$

$$u' = \frac{fc_2 - rc_1}{f^2 + r^2}e^{-rt}\times f\cos(ft) - \frac{fc_2 - rc_1}{f^2 + r^2}e^{-rt}r\sin(ft) + \frac{fc_1 + rc_2}{f^2 + r^2}e^{-rt}\times r\cos(ft) + \frac{fc_1 + rc_2}{f^2 + r^2}e^{-rt}f\sin(ft) - \frac{g'r\tan\theta}{f^2 + r^2}$$

$$u' = \cos(ft)e^{-rt}\left[f(\frac{fc_2 - rc_1}{f^2 + r^2}) + r(\frac{fc_1 + rc_2}{f^2 + r^2})\right] + \sin(ft)\times e^{-rt}\left[f\frac{(fc_1 + rc_2)}{f^2 + r^2} - r\frac{(fc_2 - rc_1)}{f^2 + r^2}\right] - \frac{g'r\tan\theta}{f^2 + r^2}$$

$$u' = \cos(ft)\times e^{-rt}\left[\frac{f^2 c_2 - rfc_1 + rfc_1 + r^2 c_2}{f^2 + r^2}\right] + \sin(ft)\times e^{-rt}\left[\frac{f^2 c_1 + frc_2 - rfc_2 + r^2 c_1}{f^2 + r^2}\right] - \frac{g'r\tan\theta}{f^2 + r^2}$$

$$u' = c_2 e^{-rt}\cos(ft) + c_1 e^{-rt}\sin(ft) - \frac{g'r\tan\theta}{f^2 + r^2} **ok$$

$$\frac{dy'}{dt} = \frac{fc_1 + rc_2}{f^2 + r^2}\left[-re^{-rt}\sin(ft) + fe^{-rt}\cos(ft)\right] + \frac{fc_2 - rc_1}{f^2 + r^2}\left[-re^{-rt}\cos(ft) - fe^{-rt}\sin(ft)\right] + \frac{g'f\tan\theta}{f^2 + r^2}$$

$$v' = e^{-rt}\sin(ft)\left[\frac{-rfc_1 - r^2c_2}{f^2 + r^2}\right] + e^{-rt}\cos(ft)\times\left[\frac{f^2c_1 + rfc_2}{f^2 + r^2}\right] + \frac{r^2c_1 - rfc_2}{f^2 + r^2}\cos(ft)\times e^{-rt} + \frac{-f^2c_2 + rfc_1}{f^2 + r^2}e^{-rt}\sin(ft) + \frac{g'f\tan\theta}{f^2 + r^2}$$

$$v' = e^{-rt}\times\sin(ft)\times -\left[\frac{f^2c_2 - rfc_1 + r^2c_2 + rfc_1}{f^2 + r^2}\right] + \cos(ft)\times e^{-rt}\left[\frac{r^2c_1 - rfc_2 + f^2c_1 + rfc_2}{f^2 + r^2}\right] + \frac{g'f\tan\theta}{f^2 + r^2}$$

$$v' = -c_2 e^{-rt}\sin(ft) + c_1 e^{-rt}\cos(ft) + \frac{g'f\tan\theta}{f^2 + r^2} \quad ***ok$$

$$\frac{du'}{dt} = -g'\tan\theta + fv - ru$$

$$-rc_1 e^{-rt}\sin(ft) + fc_1 e^{-rt}\cos(ft) - rc_2 e^{-rt}\cos(ft) - fc_2 e^{-rt}\sin(ft)$$

$$= -g'\tan\theta + fc_1 e^{-rt}\cos(ft) - fc_2 e^{-rt}\sin(ft) + \frac{g'f^2\tan\theta}{f^2 + r^2} - rc_1 e^{-rt}\sin(ft) - rc_2 e^{-rt}\cos(ft)$$

$$+ \frac{g'r^2\tan\theta}{f^2 + r^2} = 0$$

$$\frac{-g'f^2\tan\theta - g'r^2\tan\theta + g'f^2\tan\theta + g'r^2\tan\theta}{f^2 + r^2} = 0$$

$$0 = 0 \quad ***ok$$

$$\frac{dv'}{dt} = -rc_1 e^{-rt}\cos(ft) - fc_1 e^{-rt}\sin(ft) + rc_2 e^{-rt}\sin(ft) - fc_2 e^{-rt}\cos(ft)$$

$$\frac{dv'}{dt} = -fu' - rv'$$

$$-rc_1 e^{-rt}\cos(ft) - fc_1 e^{-rt}\sin(ft) + rc_2 e^{-rt}\sin(ft) - fc_2 e^{-rt}\cos(ft) = -fc_1 e^{-rt}\sin(ft)$$

$$-fc_2 e^{rt}\cos(ft) + \frac{g'rf\tan\theta}{f^2 + r^2} - rc_1 e^{-rt}\cos(ft) + rc_2 e^{-rt}\sin(ft) - \frac{g'fr\tan\theta}{f^2 + r^2}$$

$$0 = 0 \quad ***ok$$

$$u'(t=0) = c_2 - \frac{g'r\tan\theta}{f^2 + r^2} = u_p$$

$$c_2 = u_p + \frac{g'r\tan\theta}{f^2 + r^2}$$

$$v(t=0) = v_p$$

$$c_1 + \frac{g'f\tan\theta}{f^2 + r^2} = v_p$$

$$c_1 = v_p - \frac{g'f\tan\theta}{f^2 + r^2}$$

**Appendix B**

Derivation of the analytical model for calculating Qv:

[Figure]

$$if\ (x=0) = 0 \rightarrow y = 0$$

$$y = 0 \rightarrow c = 0$$

$$if\ (x = -L_1)Then$$

$$y_1 = aL_1^2 - bL_1 \rightarrow$$

$$H_1 = aL_1^2 - bL_1 \mathbf{***(1)}$$

$$if\ (x = L_2)Then$$

$$y_2 = aL_2^2 + bL_2$$

$$H_2 = aL_2^2 + bL_2 \mathbf{***(2)}$$

$$(1) \rightarrow bL_1 = aL_1^2 - H_1 \rightarrow$$

$$b = \frac{aL_1^2 - H_1}{L_1} \mathbf{***(3)}$$

$$(3),(2) \rightarrow H_2 = aL_2^2 + L_2(\frac{aL_1^2 - H_1}{L_1})$$

$$H_2 = aL_2^2 + aL_1L_2 - \frac{L_2}{L_1}H_1$$

$$H_2 + \frac{L_2}{L}H_1 = a(L_2^2 + L_1L_2) \rightarrow$$

$$a = \frac{H_2L_1 + L_2H_1}{L_1(L_1L_2 + L_2^2)} \mathbf{**(4)}$$

$$(4),(3) \rightarrow b = aL_1 - \frac{H_1}{L_1} = \frac{H_2L_1^2 + L_2L_1H_1}{L_1(L_1L_2 + L_2^2)} - \frac{H_1}{L_1}$$

$$b = \frac{H_2L_1^2 + L_2H_1L_1}{L_1(L_2^2 + L_1L_2)} - \frac{H_1}{L_1}$$

$$b = \frac{H_2L_1 + L_2H_1}{L_2^2 + L_1L_2} - \frac{H_1}{L_1} **(5)$$

$$Q_V = \int U dA$$

$$dA = \int h dx - \int y dx$$

$$h = Ae^{-\alpha x} \rightarrow H_1 = Ae^{\alpha L_1} \rightarrow A = \frac{H_1}{e^{\alpha L_1}} **(6)$$

$$if\ (x = L_2) \to H_2 = A e^{-\alpha L_2} \to H_2 = \frac{H_1}{e^{\alpha L_1}} e^{-\alpha L_2}$$

$$\frac{H_2}{H_1} = e^{-\alpha(L_1 + L_2)} \to Ln\frac{H_2}{H_1} = -\alpha(L_1 + L_2)$$

$$\alpha = -\frac{1}{L_1 + L_2} Ln\frac{H_2}{H_1} \ **(7)$$

$$\int hdx = \int_{L_1}^{L_2} A e^{-\alpha}dx = \frac{A}{-\alpha}(e^{-\alpha L_2} - e^{\alpha L_1})$$

$$\int_{L_1}^{L_2} ydx = \int_{L_1}^{L_2}(ax^2 + bx)dx = \frac{a}{3}(L_2^3 + L_1^3) + \frac{b}{2}(L_2^2 - L_1^2)$$

$$Q_V = \frac{g'}{f}\frac{H_2 - H}{L_2 - L_1}\left[\left[\frac{-A}{\alpha}(e^{-\alpha L_2} - e^{\alpha L_1})\right] - \left[\frac{a}{3}(L_2^3 + L_1^3)\right] + \frac{b}{2}(L_2^2 - L_1^2)\right]$$

$$\alpha = \frac{H_2 L_1 + H_1 L_2}{L_1(L_1 L_2 + L_2^2)}$$

$$b = \frac{H_2 L_1 + L_2 H_1}{L_2^2 + L_1 L_2} - \frac{H_1}{L_1}$$

$$A = \frac{H_1}{e^{\alpha L_1}}$$

$$\alpha = \frac{-1}{L_1 + L_2} Ln\frac{H_2}{H_1}$$

$$if \to |L_1| \cong |L_2| \cong L$$

$$Q_V = \frac{g'}{f} \frac{H_2 - H_1}{L_2 - L_1} \left[ \left[ \frac{-A}{\alpha} (e^{-\alpha L} - e^{\alpha L}) \right] - \left[ \frac{2a}{3} (L^3) \right] \right]$$

$$Q_V = \frac{g'}{f} \frac{H_2 - H}{L_2 - L_1} \left[ \left[ \frac{2A}{\alpha} (\frac{e^{\alpha L} - e^{-\alpha L}}{2}) \right] - \left[ \frac{2a}{3} (L^3) \right] \right]$$

$$Q_V = \frac{g'}{f} \frac{H_2 - H_1}{L_2 - L_1} \left[ \frac{2A}{\alpha} Sinh(\alpha L) - \frac{2}{3} \alpha L^3 \right]$$

Or

$$Q_V = \frac{g'}{f} \frac{H_2 - H_1}{2L} \left[ \frac{2A}{\alpha} Sinh(\alpha L) - \frac{2}{3} \alpha L^3 \right]$$

Where

$$\alpha = \frac{H_2 + H_1}{2L_2^2}$$

$$b = \frac{H_2 - H_1}{2L}$$

$$A = \frac{H_1}{e^{\alpha L}}$$

$$\alpha = \frac{-1}{L_1 + L_2} Ln \frac{H_2}{H_1}$$